# The effects of zinc sulfate on mycelial enzyme activity and metabolites of *Pholiota adiposa*

**Xiao-ying Ma, Tao Yang, Jun Xiao, Peng Zhang\***

The Edible Fungus Institute, Liaoning Academy of Agricultural Sciences, Shenyang, Liaoning, China

\* 59860651@qq.com

## Abstract

The aim of this study was to investigate the effect of zinc sulphate on the activities of different enzymes and metabolites of *Pholiota adiposa*. In the experiment, we used the conventional enzyme activity assay to determine the changes of six indicators, including protein content, laccase activity, cellulase activity, amylase activity and polyphenol oxidase activity, under different concentrations of zinc sulphate treatment. The results showed that the activities of amylase, laccase, cellulase and peroxidase were $Zn^{2+}(200)>Zn^{2+}(0)>Zn^{2+}(400)>Zn^{2+}(800)$. The activities of catalase and superoxide dismutase were $Zn^{2+}(200)>Zn^{2+}(400)>Zn^{2+}(800)$, and zinc sulfate could significantly affect the activity of polylipic squamase in a dose-dependent manner. Further correlation analysis showed that all six enzyme activities were significantly correlated with each other ($P<001$); the results of the statistical model test showed that the regression model constructed was statistically significant; overall the residuals met the conditions of normal distribution, and the corresponding points of different enzyme activities Q—Q' were more evenly distributed around $y = x$, and all fell in the 90% acceptance interval, thus the series was considered to obey normal distribution; the results of the principal The results of the principal component analysis showed that principal component 1 was positively correlated with amylase, laccase and cellulase. Principal component 2 was positively correlated with superoxide dismutase and catalase, and negatively correlated with peroxidase. The analysis of Metabonomic data revealed that zinc sulfate had a significant impact on the expression of metabolites in the mycelium. Moreover, varying concentrations of zinc sulfate exerted significant effects on the levels of amino acids, organic acids, and gluconic acid. This conclusion was confirmed by other experimental data. The results of the study provide a scientific reference for better research, development and utilization of *Pholiota adiposa*.

## Introduction

*Pholiota adiposa*, belonging to the Fungi genus, is a widely distributed edible fungus with significant industrial importance. The fruiting body of *Pholiota adiposa* has a fan-shaped morphology, fleshy texture, and a soft, yellowish-brown color [1]. *Pholiota adiposa* is a nutritious and delicious edible fungus, characterized by its high protein content, low fat and sugar

**Data Availability Statement:** All relevant data are within the paper and its Supporting Information files.

**Funding:** the Discipline Construction Plan of Liaoning Province Academy of Agricultural

Sciences (Grant No. 2022DD072311). The funders and correspondence Zhang Peng designed and reviewed this manuscript.

**Competing interests:** The authors report no potential conflicts of interest.

content, and abundant potassium and trace elements [2]. Additionally, *Pholiota adiposa* exhibits hypoglycemic, immune regulatory, anti-tumor, and intestinal flora regulating effects [3–6].

Zinc sulfate (ZnSO4) is an inorganic compound that is soluble in water. It has widespread applications in medicine, agriculture, industry, and various other fields. In agriculture, zinc sulfate serves as a trace element fertilizer, enhancing crop yield and quality. Moreover, it is commonly used in the production of edible fungi, promoting growth, nutrient synthesis, and disease prevention. Various concentrations of zinc sulfate have been found to significantly increase the yield of *Flammulina velutipes* and the content of polysaccharides and zinc in its fruiting body [7]. Additionally, zinc supplementation has been shown to increase the ash and protein content of *Pleurotus ostreatus*, as well as improve its scavenging activity of DPPH free radicals (96.3%) [8]. Furthermore, the addition of zinc has been found to enhance the yield and antioxidant capacity of *Lentinus edodes* [9]. Interestingly, low concentrations of zinc can promote the growth of sulfur mycelium. However, with increasing zinc concentration, the changes in mycelial soluble protein and polysaccharide content first increase and then decrease, whereas the changes in soluble sugar content exhibit the opposite trend [10]. Research has indicated a correlation between enzyme activity, strain activity, and zinc concentration [11]. In *Morchella*, zinc plays a crucial role in the biosynthesis of nucleoprotein, nucleic acid, and other compounds, greatly influencing cell membrane structure and function [12].

In the process of growth and development, edible fungi synthesize and secrete extracellular enzymes to obtain nutrients necessary for their own growth. The activity of these enzymes affects the absorption and utilization rate of nutrients by edible fungi [13]. Furthermore, it determines the yield and quality of edible fungi. The mycelium of edible fungi degrades cellulose into monosaccharides through enzyme catalysis, providing nutrition for self-extension [14]. By examining the extracellular enzyme activity of edible fungi under different conditions, we can understand the decomposition degree and activity, and create an optimal environment for the growth of hyphae and sporangia [15]. Researchers investigated the changes in extracellular enzyme activity and fruiting body development at different stages of the *Lentinus edodes* growth cycle using spectroscopy. They observed a significant relationship between the change in extracellular enzyme activity and the growth of *Lentinus edodes* [16]. Similarly, the changes in extracellular enzyme activity of *Pleurotus eryngii* were studied under environmental stress. It was found that the mycelium of *Pleurotus eryngii* could secrete these enzymes to maintain active oxygen metabolism balance and protect the membrane structure, allowing the fruiting body to withstand stress to a certain extent [17]. Different strains have different abilities in lignin degradation, and it was discovered that laccase activity correlates positively with the rate of lignin degradation [18]. The metabolome refers to a collection of small molecular compounds that participate in an organism's metabolism, aiding in normal growth, function, and development [19]. Currently, metabonomics is used in various aspects of edible fungi research [20–22].

The present study aimed to investigate the impact of zinc sulfate on the growth of *Pholiota adiposa* utilizing enzymology and metabolomics techniques. The aim is to identify the concentration that indicates high activity and analyze the correlation between them. Regression analysis, principal component analysis, and evaluation scores are conducted to provide a reference basis for optimal culture conditions and utilization of edible fungi.

## Materials and methods

### Materials and chemicals

*Pholiota adiposa* was selected by the Breeding Laboratory of the Institute of Edible Fungi at the Liaoning Academy of Agricultural Sciences; Glucose, peptone, Agar, potassium dihydrogen

phosphate, sodium dihydrogen phosphate, and zinc sulfate are all analytically pure, that were purchased from National Pharmaceutical Group Chemical Reagent Co., Ltd; HH-3A single-row single-control three-hole water bath pot is purchased from Changzhou Guoyu Instrument Manufacturing Co., Ltd; Electronic balance is purchased from Mettler-Toledo Instrument Co., Ltd; Ultrasonic cleaner is purchased from Kunshan Ultrasonic Instrument Co., Ltd; HZQ-QX full-temperature oscillator is purchased from Harbin Donglian Electronic Technology Development Co., Ltd; Dry heat disinfection box is purchased from Shanghai Jinghong Experimental Equipment Co., Ltd; The A11 high-speed pulverizer was purchased from IKA Group in Germany, the ultra-pure water system in Cascada laboratory was purchased from Pall Corporation, and the enzyme-labeled analyzer Infinite F50 was purchased from Tecan in Switzerland; The activities of amylase, laccase, cellulase, catalase, peroxidase, and superoxide dismutase were determined using a kit provided by Solebo Biological Reagent Co., Ltd., and the assays were performed according to the instructions.

## Preparation of *Pholiota adiposa* mycelium

200 mg of zinc sulfate was prepared, mixed with 400 mg of PDA medium, sterilized under high pressure, and poured into a 9mm Petri dish. After the mixture cooled and solidified, the activated bulbs were inoculated into individual Petri dishes containing 1 clot each. The dishes were then cultured at 25 degrees Celsius for 5–7 days, with 20 repetitions in each treatment.

## Preparation of samples

The hyphae were treated with various concentrations of zinc sulfate. Each sample was scraped off and weighed 1 g, then transferred into a 50 mL centrifuge tube. The hyphae was ground and broken in ice bath aseptic water at a ratio of 1:30. The resulting homogenate was collected and the supernatant was separated by centrifugation. The supernatant was stored at 4°C for further analysis. It was utilized to measure the activity levels of laccase, cellulase, amylase, peroxidase, catalase, and superoxide dismutase.

## Determination of protein content

The protein content was determined using the BCA method and a related kit. The specific steps carried out for this analysis involved using the BCA protein content determination kit provided by Solebo Biological Reagent Co., Ltd.

## Determination of enzyme activity

The kit utilized in this study employed a double antibody one-step sandwich enzyme-linked immunosorbent assay (ELISA). The procedure involved adding samples, standards, and HRP-labeled antibodies to micropores coated with enzyme antibodies. Subsequently, the mixture was incubated and thoroughly washed. The color development was then achieved by employing the substrate TMB. Under the catalysis of peroxidase, TMB was converted to blue and finally to yellow under the influence of acid. The intensity of the resulting color was directly proportional to the amount of the target enzymes present in the sample. The activities of amylase, laccase, cellulase, catalase, peroxidase, and superoxide dismutase were determined using the kit provided by Solebo Biological Reagent Co., Ltd. All procedures were conducted as instructed in the kit manual.

## Metabonomic analysis

**Metabonomics sample processing.**    The collected samples were thawed on ice, and metabolite were extracted with 50% methanol. Briefly, 100 mg of sample was extracted with 1 ml of precooled 50% methanol, vortexed for 1 min, and incubated at room temperature for 10 min; the extraction mixture was then stored overnight at -20˚C. After centrifugation at 4,000 g for 20 min, the supernatants were transferred into new 96-well plates. The samples were stored at -80˚ C prior to the LC-MS analysis. In addition, pooled QC samples were also prepared by combining 10 μ L of each extraction mixture [23].

**Instrument parameters.**    A high-resolution tandem mass spectrometer Q-Exactive (Thermo Scientific) was used to detect metabolites eluted form the column. The Q-Exactive was operated in both positive and negative ion modes. Precursor spectra (70–1050 m/z) were collected at 70,000 resolution to hit an AGC target of 3e6. The maximum inject time was set to 100 ms. A top 3 configuration to acquire data was set in DDA mode. Fragment spectra were collected at 17,500 resolution to hit an AGC target of 1e5 with a maximum inject time of 80 ms. In order to evaluate the stability of the LC-MS during the whole acquisition, a quality control sample (Pool of all samples) was acquired after every 10 samples [24, 25].

**Data processing.**    The acquired MS data pretreatments including peak picking, peak grouping, retention time correction, second peak grouping, and annotation of isotopes and adducts was performed using XCMS software. LC−MS raw data files were converted into mzXML format and then processed by the XCMS, CAMERA and metaX toolbox implemented with the R software. Each ion was identified by combining retention time (RT) and m/z data. Intensities of each peaks were recorded and a three dimensional matrix containing arbitrarily assigned peak indices (retention time-m/z pairs), sample names (observations) and ion intensity information (variables) was generated. The online KEGG, HMDB database was used to annotate the metabolites by matching the exact molecular mass data (m/z) of samples with those from database. If a mass difference between observed and the database value was less than 10 ppm, the metabolite would be annotated and the molecular formula of metabolites would further be identified and validated by the isotopic distribution measurements. We also used a in-house fragment spectrum library of metabolites to validate the metabolite identidification [26].

## Statistical analysis

All experiments were repeated three times. Statistical analysis was performed using SPSS 20.0 software and the graphs were generated with GraphPad Prism 8.0.2.

# Results

## Protein determination

The total protein was determined using the BCA method with a relevant kit. The protein content of the different treatments was determined using a protein standard curve (Fig 1) with the equation Y = 0.1596x-0.0003131.The results indicated (Fig 2) that the protein content followed the order: Zn2+ (0) > 400 > 200 > 800.The difference was found to be statistically significant and highly significant using one-way analysis of variance (ANOVA) and the LSD test in SPSS 20.0, respectively. The results indicated no significant difference in protein content among the different treatments, with no statistical significance observed.

## Enzyme activity determination

According to the standard curve of six enzyme activity (Fig 3), we obtained the enzyme activity of different treatment groups (Fig 4). According to the results of ANOVA analysis, the enzyme

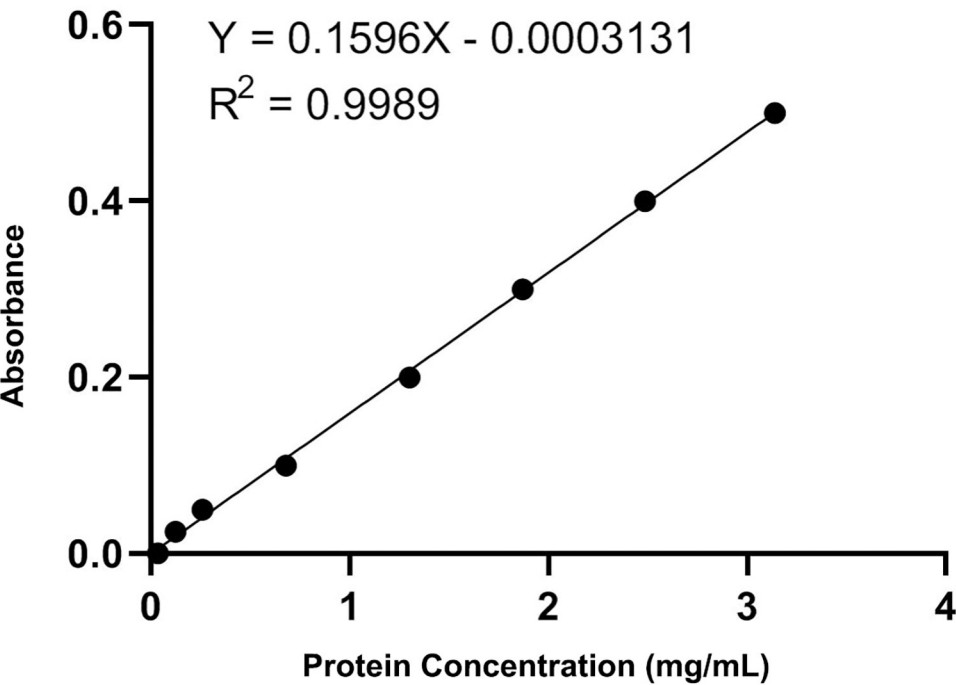

**Fig 1. Protein standard curve.**

activity P values of different treatment groups were all less than 0.001, which was statistically significant (Table 1). The activities of laccase, catalase, peroxidase, and superoxide dismutase in the Zn+ group exhibited significant differences (* P < 0.001). The activities of amylase and cellulase were also significantly different (P < 0.05). A heat map was constructed based on the results of the enzyme activity (Fig 5). The analysis demonstrated that the three replicates for each treatment could be classified into distinct categories, indicating the accuracy of the determination results. Catalase and superoxide dismutase results were grouped together, amylase, laccase, and cellulase were classified into another group, and peroxidase was placed into a separate category.

## Correlation analysis

Table 2 presents the results of the correlation analysis conducted on the mycelial enzyme activity of *Pholiota adiposa*, which was treated with various concentrations of zinc. Amylase exhibited a significant positive correlation with laccase, cellulase, and catalase(P < 0.001)., and also displayed a positive correlation with superoxide dismutase (P < 0.05).Laccase demonstrated positive correlations with cellulase and catalase(P<0.0 1), while displaying a negative correlation with peroxidase(P<0.0 1).Cellulase showed positive correlations with amylase, laccase, and catalase(P<0.0 1). Conversely, peroxidase exhibited a negative correlation with catalase and superoxide dismutase (P < 0.001), but displayed positive correlations with other enzyme activities (P < 0.01) (Fig 6).

## Linear regression analysis

Linear regression analysis was conducted on the results of mycelial enzyme activity of *Pholiota adiposa* treated with various concentrations of ZnSO4 to establish the relationship between ZnSO4 concentration and enzyme activity. The results (Table 3) demonstrated that the

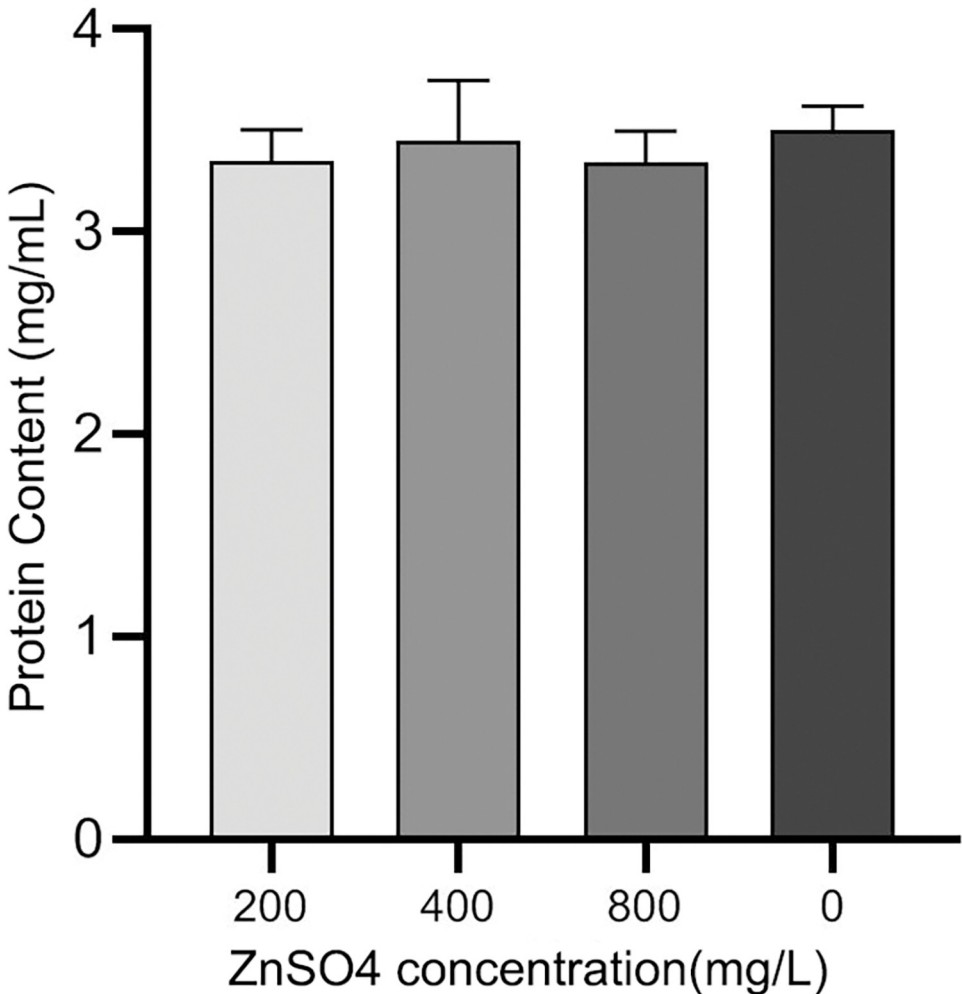

**Fig 2. The effect of different concentrations of ZnSO4 on protein content.**

multiple correlation coefficient (r) was 0.990.The determination coefficient (R2), with a value of 0.979, was used to assess the accuracy of the model. This indicates that the concentration of ZnSO4 can predict the activity of superoxide dismutase, cellulase, laccase, amylase, peroxidase, and catalase in lipid scale *Pholiota adiposa* with an accuracy of 97.9%.The statistical test of the model using analysis of variance (ANOVA) is presented in Table 4. The test yielded a Franch value of 39.768, with a p-value of 0.000, which indicates that the regression model is statistically significant. Fig 7 illustrates that the overall residual adheres to the assumption of normal distribution. Analysis of Fig 8 reveals that the points representing different enzyme activities (QmurQ') are uniformly distributed in close proximity to yyogx, all falling within the 90% confidence interval. Therefore, it can be inferred that the sequence follows a normal distribution.

## Principal component analysis

The principal component analysis was conducted using the six indexes of lipid scale *Pholiota adiposa* enzyme activity determined earlier. The principal component analysis was conducted using the six indexes of lipid scale *Pholiota adiposa* enzyme activity determined earlier. The

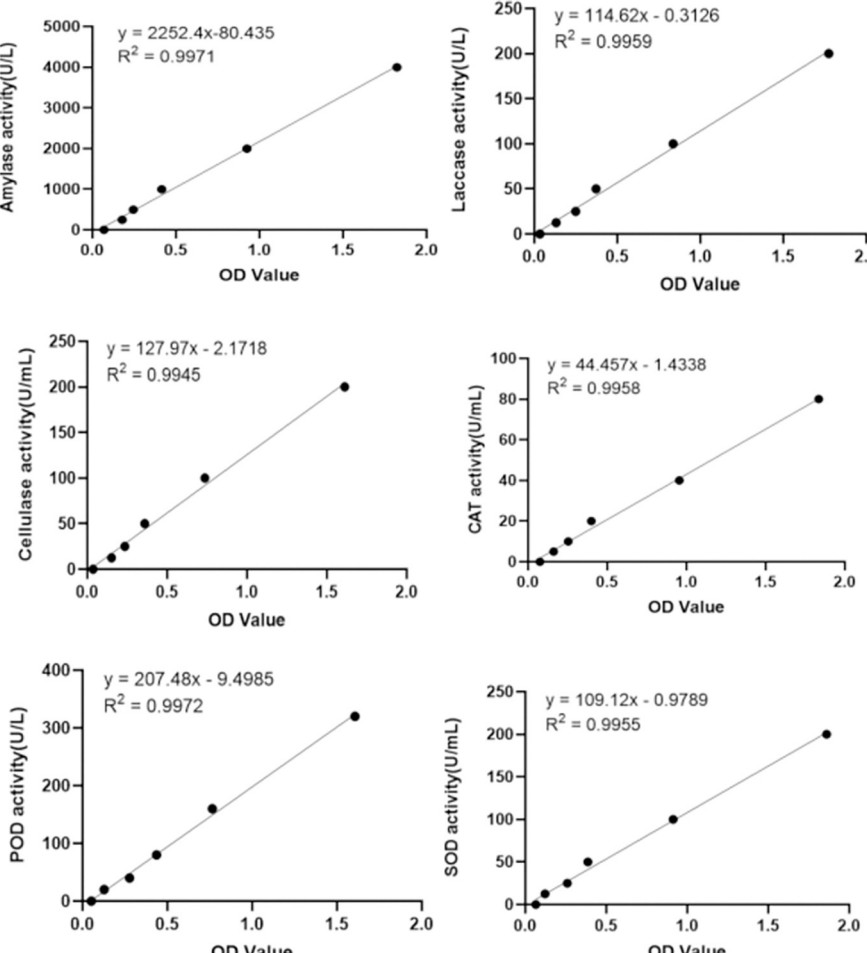

**Fig 3. The standard curve of different enzyme activities.**

Kaiser-Meyer-Olkin (KMO) test showed that KMO > 0.5 and P < 0.05 (Table 5).This index is appropriate for principal component analysis. The scree plot (Fig 9) revealed one factor with an eigenvalue greater than 1, but for enhanced concentration factors, two factors were selected. Component 1 makes the largest contribution to explaining the original variable, while the characteristic root value of the second factor is less than 1 and can be disregarded (Fig 10). The analysis from Table 6 reveals that the first principal component has the largest variance contribution rate (82.154%) and eigenvalue (4.929), which plays a significant role in the activity of the lipid scale *Pholiota adiposa* enzyme. The second principal component has a variance contribution rate of 15.725% and an eigenvalue of 0.944.The cumulative variance contribution rate of the two principal components reached 97.879%, encompassing the information of the main components.

The component matrix (Table 7) indicated that amylase, laccase, and cellulase have a significant impact on the first principal component. The load diagram revealed that catalase and superoxide dismutase were clustered together, while amylase, laccase, and cellulase formed another cluster. Cellulase had the highest load on component 1, while superoxide dismutase exhibited the highest load on component 2.Transforming it into a feature vector allows obtaining the expressions of the two principal components. The rotational component matrix

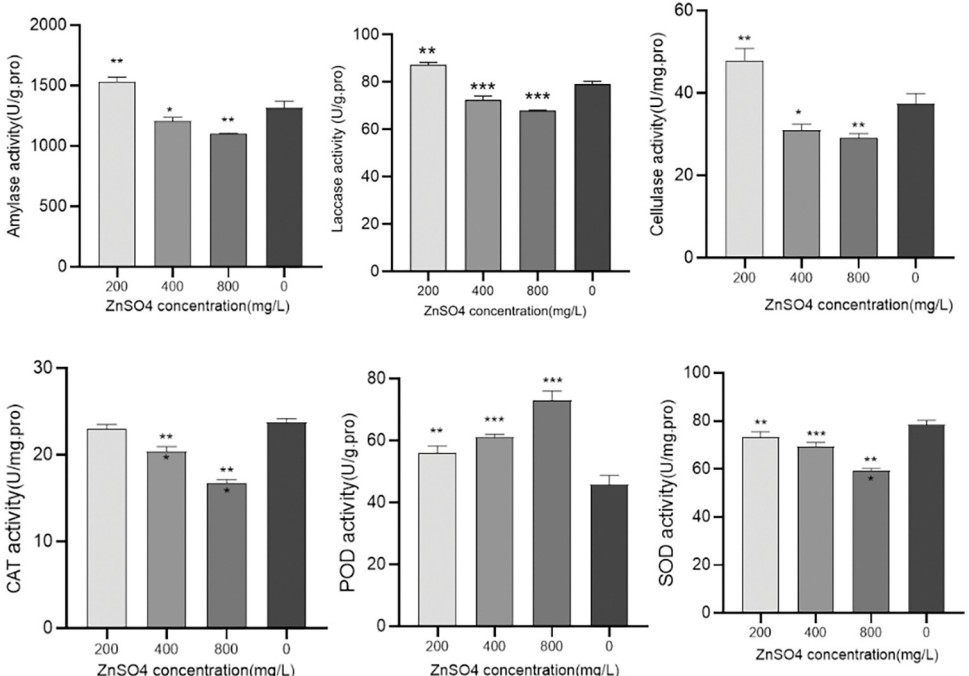

**Fig 4. Effects of different concentrations of ZnSO4 on the activities of six enzymes in the mycelia of *Pholiota adiposa* ($*P<0.05, **P<0.01, ***P<0.001$).**

(Table 8) showed a significant positive correlation between principal component 1 and amylase, laccase, and cellulase, whereas principal component 2 exhibited positive correlations with superoxide dismutase and catalase, and a negative correlation with peroxidase.

$$Z_1 = 0.43X_1+0.43X_2+0.41X_3-0.4X_4+0.39X_5-0.39X_6$$

$$Z_2 = -0.25X_1+0.3X_2+0.39X_3-0.45X_4+0.5X_5+0.5X_6$$

The enzymes amylase, laccase, cellulase, catalase, peroxidase, and superoxide dismutase are represented by x1, x2, x3. . . X6 respectively. The standardized values of the six indexes were used to calculate the principal component scores Z1 and Z2. The comprehensive evaluation score K value of each treatment was obtained by considering the weight of the variance contribution rate of the two factors, and then transformed into the percentile system. According to Table 9, the highest comprehensive score was 89.8 for $Zn^{2+}(200)$, followed by $Zn^{2+}(0)$, $Zn^{2+}(400)$, $Zn^{2+}(800)$. This suggests that the concentration of $Zn^{2+}(200)$ had the most significant impact on the enzyme activity (Table 9).

**Table 1. Analysis of variance (ANOVA) for enzyme activity of *Pholiota adiposa* in different treatments.**

|  | Sum of square | df | MS | F value | P value |
|---|---|---|---|---|---|
| Amylase | .311 | 3 | .104 | 80.651 | .000 |
| Laccase | .001 | 3 | .000 | 168.622 | .000 |
| Cellulase | 655.155 | 3 | 218.385 | 50.582 | .000 |
| CAT | 91.614 | 3 | 30.538 | 129.860 | .000 |
| POD | .001 | 3 | .000 | 61.121 | .000 |
| SOD | 581.466 | 3 | 193.822 | 61.149 | .000 |

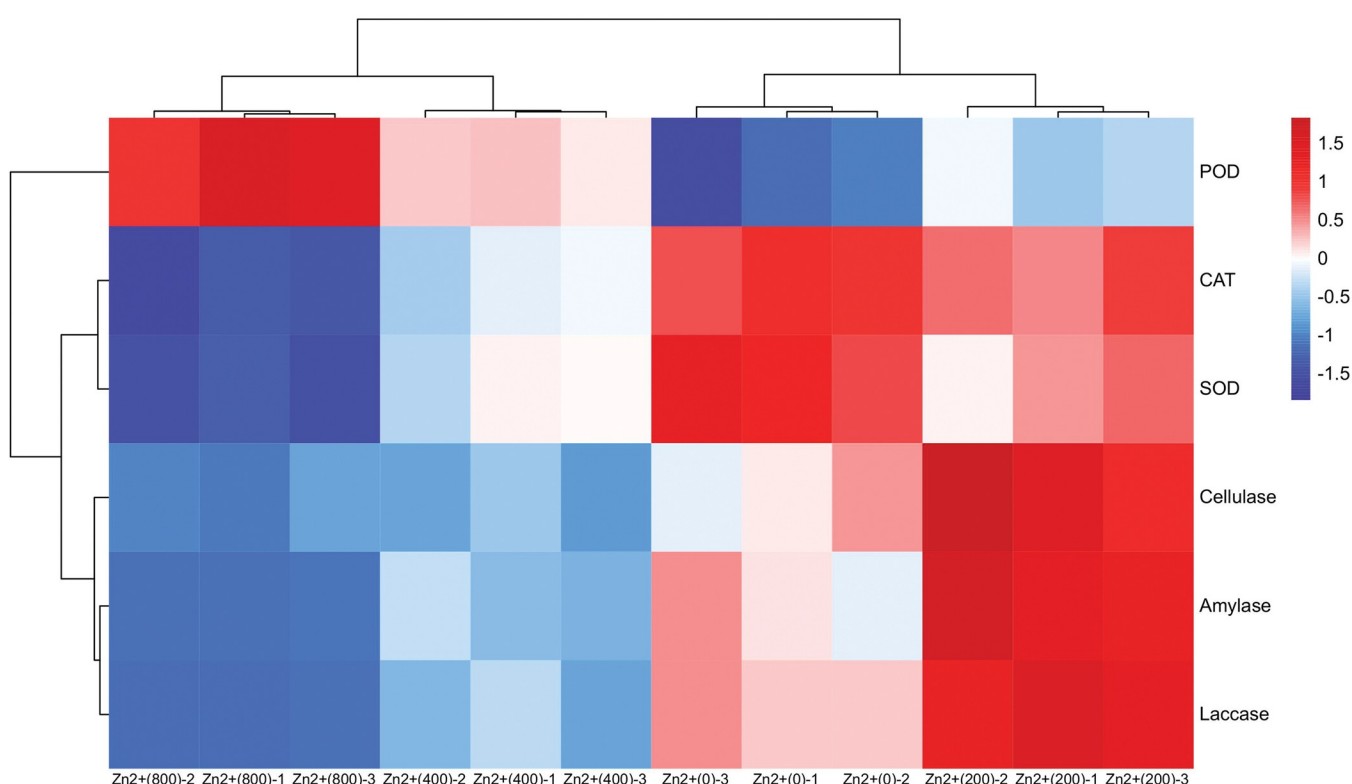

**Fig 5. Heat map illustrating the effect of varying concentrations of ZnSO4 on the activities of six enzymes in *Pholiota adiposa* mycelium.**

## B. Investigation of ZnSO$_4$ Metabolism in *Pholiota adiposa*

**Metabolic profile analysis.** The typical ion flow diagram of the sample is shown in Fig 11A, which represents the QC sample.To assess the stability and repeatability of the system, all samples were normalized and analyzed using principal component analysis (PCA). Fig 11B shows that all QC samples are relatively aggregated. In this analysis, we utilized the XCMS software to align the peaks of the mass spectrometry data, enabling a more accurate quantification of the metabolites.The graph plots the retention time on the x-axis (Abscissa) and the mUnix z on the y-axis (ordinate).Each dot on the graph represents a specific substance, and the color of the dot indicates the density of that substance in the area. A darker color corresponds to a higher number of feature numbers.The results indicate the stability and reliability of the gas phase-mass spectrometry system throughout the analysis process. Fig 11C demonstrated a significant difference in the metabolic spectrum between the control group and the groups

**Table 2. Correlation analysis of enzyme activity of *Pholiota adiposa* in different treatments.**

|  |  | Amylase | Laccase | Cellulase | CAT | POD | SOD |
|---|---|---|---|---|---|---|---|
| Correlation | Amylase | 1.000 | 0.98 | 0.95 | 0.77 | -0.61 | 0.65 |
|  | Laccase | 0.98 | 1.000 | 0.95 | 0.82 | -0.68 | 0,72 |
|  | Cellulase | 0.95 | 0.95 | 1.000 | 0.72 | -0.51 | 0.56 |
|  | CAT | 0.77 | 0.82 | 0.72 | 1.000 | -0.92 | 0.96 |
|  | POD | -0.61 | -0.68 | -0.51 | -0.92 | 1.000 | -0.97 |
|  | SOD | 0.65 | 0,72 | 0.56 | 0.96 | -0.97 | 1.000 |

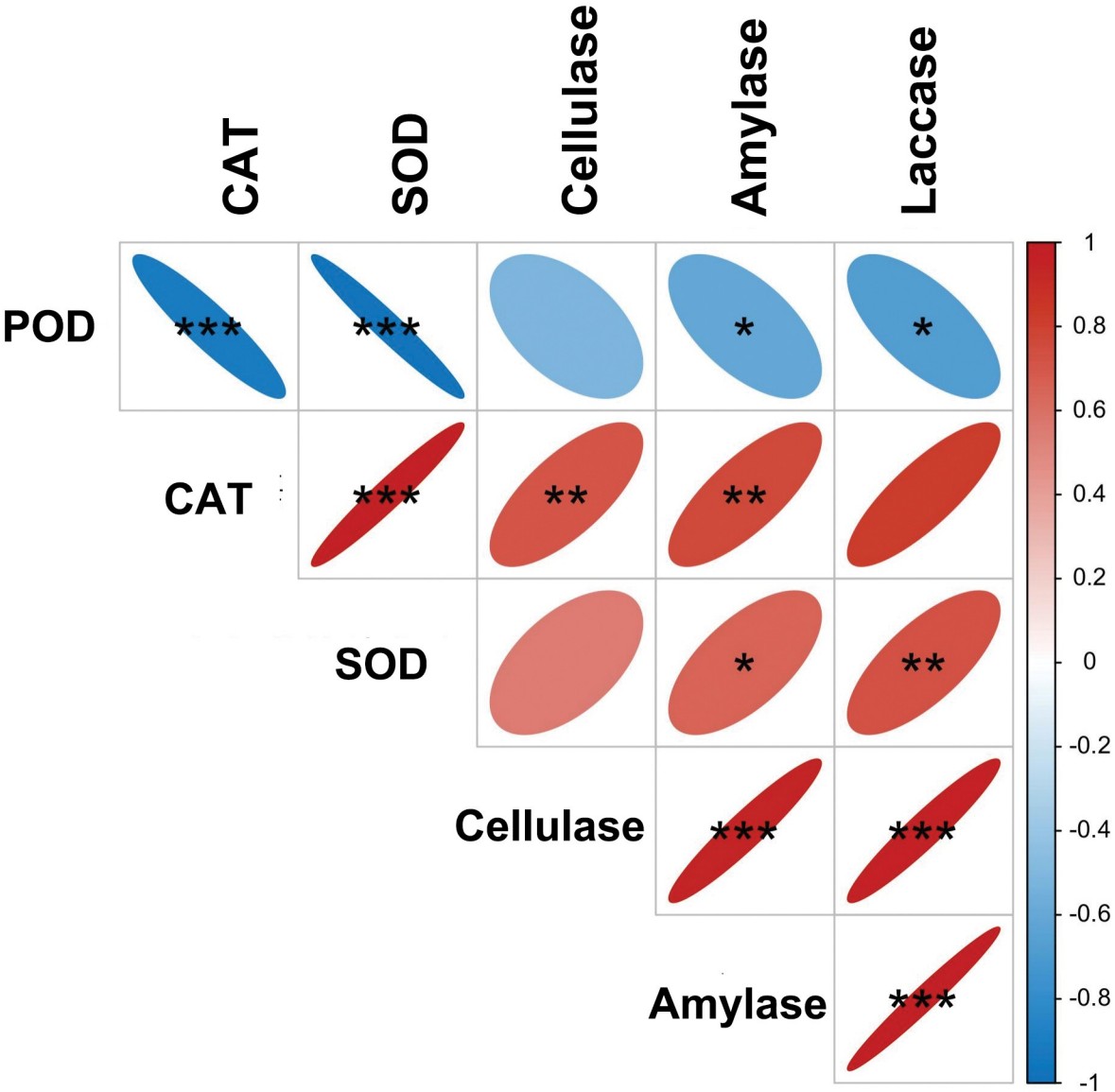

**Fig 6. Correlation matrix of different enzyme activities.**

treated with low and high concentrations of zinc sulfate.The results indicated that zinc sulfate can regulate the metabolic spectrum of *Pholiota adiposa* at a specific concentration.

**Differential metabolite analysis.**   To elucidate the overall metabolic differences induced by zinc sulfate on *Pholiota adiposa*, the P-value and fold change (FC) were calculated for each

**Table 3. Model Summaryb.**

| Model | R | R Square | Adjudted R Square | Std.Error of the Estimate | Durbin-Watson |
|---|---|---|---|---|---|
| 1 | .990[a] | .979 | .955 | .24814 | 1.500 |

a. Predictors:(constant), SOD, cellulase, laccase, amylase, POD, CAT

b. Dependent variable: ZnSO4 concentration treatment

**Table 4. Anovab.**

| Model | | Sum of Squares | df | Mean Square | F | Sig. |
|---|---|---|---|---|---|---|
| 1 | Regression | 14.692 | 6 | 2.449 | 39.768 | .000[a] |
| | Residual | .308 | 5 | .062 | | |
| | Total | 15.000 | 11 | | | |

a. Predictors:(constant), SOD, cellulase, laccase, amylase, POD, CAT

b. Dependent variable: ZnSO4 concentration treatment

concentration group of zinc sulfate compared to the control group. Differential metabolites were identified based on the criteria of $P < 0.001$ and $FC \geq 2$ or $FC \leq 0.5$ (Fig 12A). Supplementation of various concentrations of zinc sulfate resulted in the identification of 40 differential metabolites compared to the control group. Subsequently, we analyzed the change trends of differential metabolites in *Pholiota adiposa* following zinc sulfate supplementation to investigate its impact on *Pholiota adiposa* metabolism. Fig 12B displays the specific change trends of all differential metabolites in both the control group and each concentration group of zinc sulfate. The results demonstrated that a high concentration of zinc resulted in a decrease in the levels of succinic acid and tyrosine (Fig 12C), which are intermediates of the tricarboxylic acid (TCA) cycle, while increasing the secretion of organic acids by *Pholiota adiposa*.Optimal

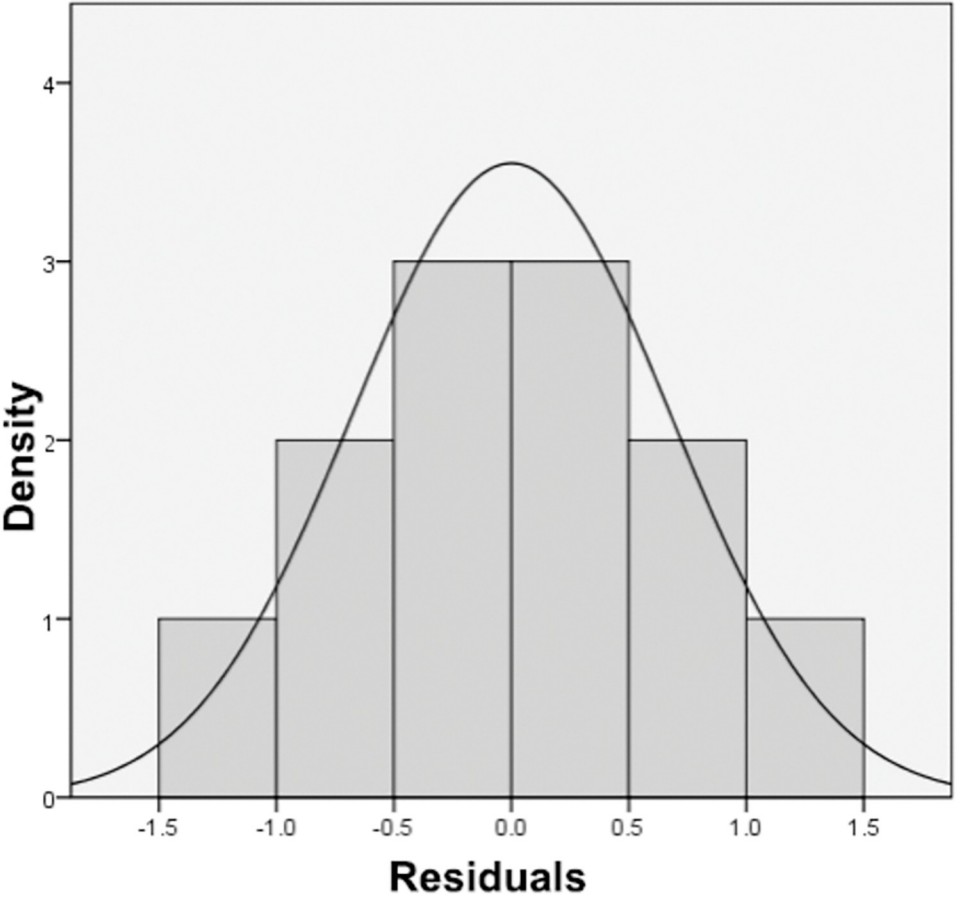

**Fig 7. Histogram of residuals.**

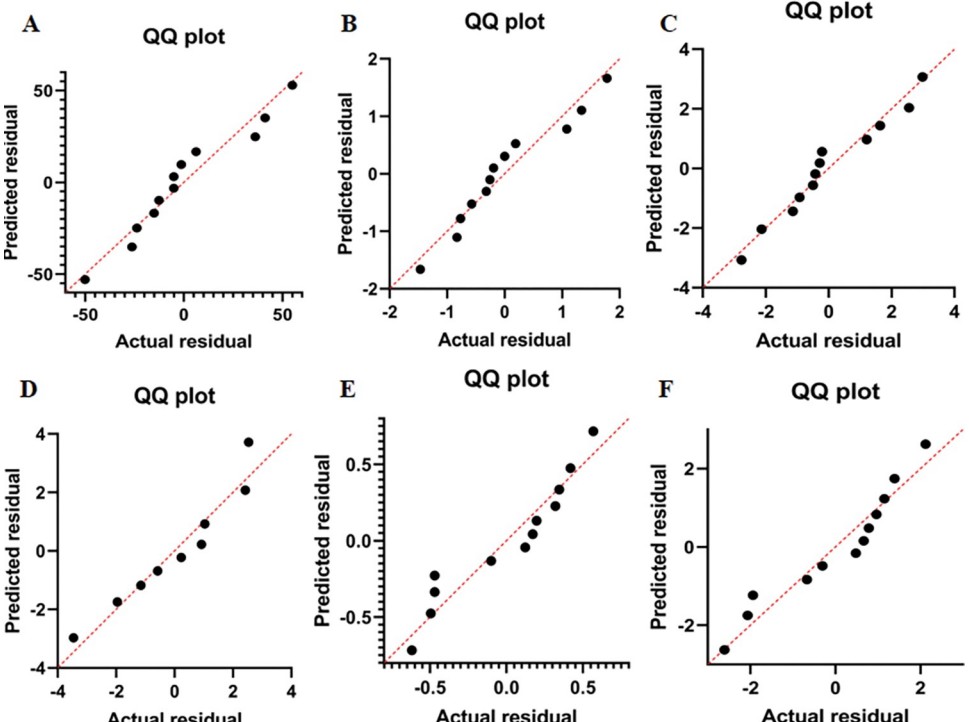

**Fig 8.** Residual QQ Plot Diagram of Different Enzyme Activities (A:Amylase; B:Laccase; C:Cellulase; D: POD; E: CAT; F:SOD).

concentrations of organic acids promoted mycelium growth, while excessive concentrations inhibited it.At high concentrations of zinc, the levels of arginine and glutamic acid increased (Fig 12D), indicating a universal osmotic protective effect of basic amino acids in *Pholiota adiposa*.

## Discussion

Edible mushrooms serve as a valuable source of dietary protein, vitamins, and minerals and are consumed widely across various cultures. Additionally, mushrooms are significant sources of enzymes, such as proteases, lipases, and cellulases, which play a crucial role in the breakdown of macromolecules and absorption of nutrients. Zinc, a necessary micronutrient for the growth and development of fungi, also contributes to the regulation of enzyme function. However, the specific impact of zinc sulfate on enzyme activity in edible fungi remains poorly characterized. Zinc, being an essential micronutrient for the growth and development of organisms including fungi, deficiency in zinc may lead to a decline in enzyme activity and impairment of metabolic function.

The mycelium's growth rate in edible fungi is closely linked to the activity of extracellular enzymes. A higher enzyme activity leads to increased absorption of nutrients from the

**Table 5. KMO and bartlett test.**

| Kaiser-Meyer-Olkin Measure of Sampling Adequacy | | .523 |
|---|---|---|
| Bartlett's Test of Sphericity | Appropriate Chi-square | 123.765 |
| | df | 15 |
| | Sig. | .000 |

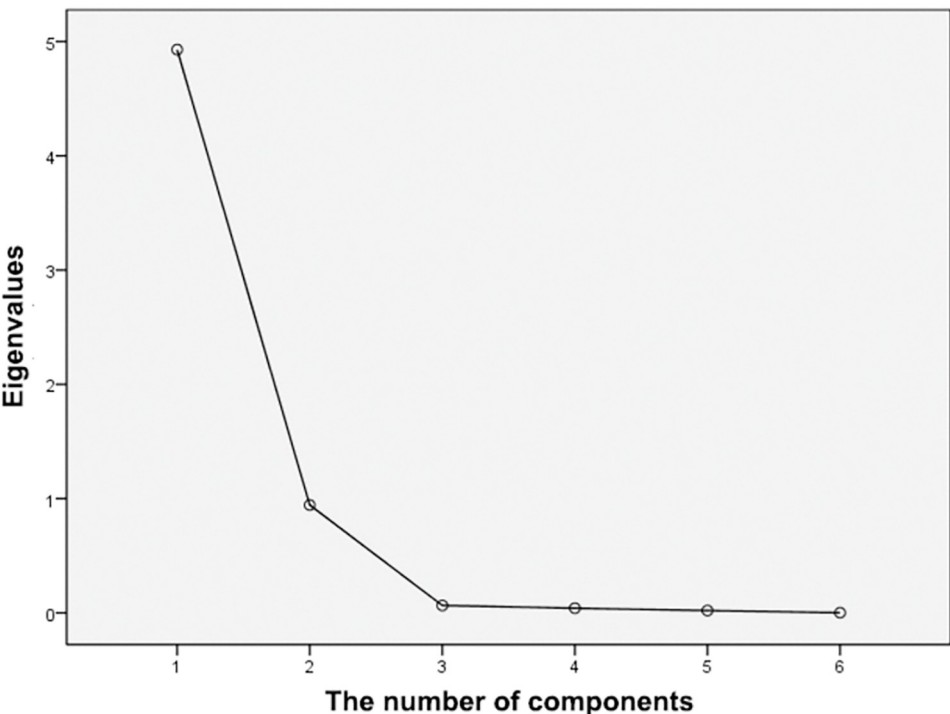

**Fig 9. The principal components analysis of gravel map.**

decomposing base material, resulting in better mycelium growth and improved agronomic characteristics of the fruiting bodies. Amylase is a general term for enzymes that hydrolyze starch or glycogen. These enzymes primarily act on soluble starch, amylose, and other substances, participating in the pathway of starch and sugar metabolism by breaking glycosidic bonds. The cellulase found in mycelium cells can decompose plant fiber into simple organic matter, which can be transformed and utilized. Cellulase also serves as a test index for strain activity [27]. During Morchella's growth and development, it secretes cellulase to hydrolyze cellulose into small glucose molecules, which are then absorbed and utilized by the Morchella mycelium and fruiting body. The activity of cellulase can reflect the physiological and biochemical activity of hyphae and fruiting body [28]. Edible fungi secrete laccase and other polyphenol oxidases to degrade lignin, producing small molecular nutrients and providing nutrition to promote mycelial growth [29, 30]. The lignin enzyme system includes primarily peroxidase and laccase [31]. Laccase is a typical polyphenol oxidase [32, 33], containing four copper ions and belonging to the copper blue oxidase family. Laccase can rapidly degrade aromatic macromolecular compounds, such as lignin, and can also contribute to the electron transport pathway during respiration. This energy contribution aids in the synthesis, transport, and accumulation of edible fungi mycelium, promoting mycelium extension and regulating fruiting body growth [34, 35]. The addition of zinc sulfate to the formula significantly boosted laccase activity in the original mycelium of apricot abalone 6 [36]. Edible fungi also secrete amylase to degrade complex macromolecules such as glycogen, dextrin, and hydrolyzed starch, producing small molecular monosaccharides that provide nutrition for mycelium growth. There is a certain correlation between amylase activity and mycelial growth rate during the growth period of edible fungi [20, 37–39].

In this study, we compared the activities of six enzymes in the mycelia of *Pholiota adiposa* cultured with varying concentrations of zinc sulfate. Based on the results of enzyme activity

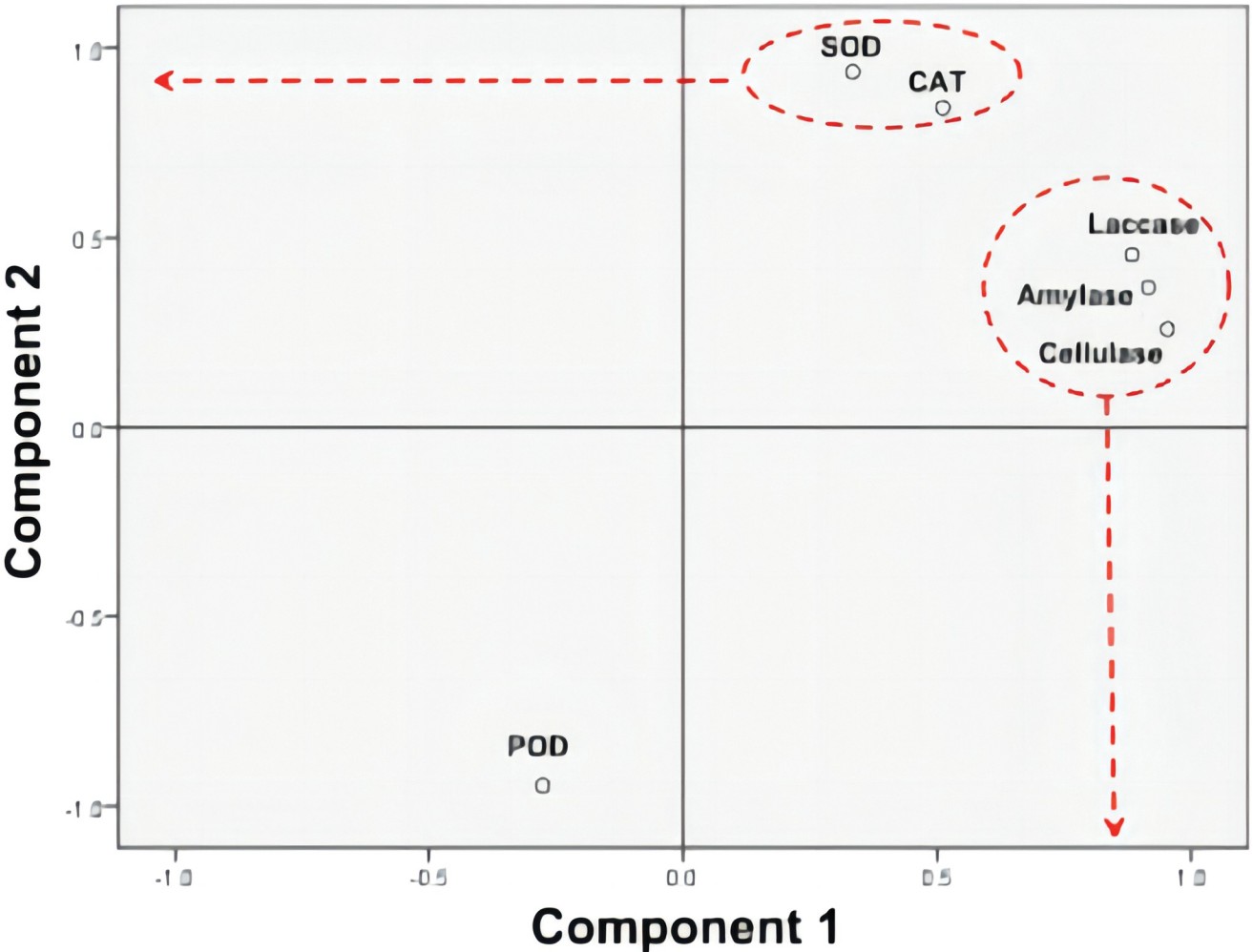

**Fig 10. Principal component analysis loading diagram.**

determination, catalase and superoxide dismutase were grouped together, while amylase, laccase, and cellulase were grouped together, and peroxidase was classified separately. Analysis of the rotational composition revealed that factor 1 included amylase, laccase, and cellulase, while factor 2 included superoxide dismutase and catalase. The results of principal component

**Table 6. Total variance of principal components interpretation.**

| Components | Initial eigenvalue | | | Extraction sums of squared loadings | | | Rotated sum of squares loadings | | |
|---|---|---|---|---|---|---|---|---|---|
| | Total | of Variance % | Cumulative % | Total | of Variance % | Cumulative % | Total | of Variance % | Cumulative % |
| 1 | 4.929 | 82.154 | 82.154 | 4.929 | 82.154 | 82.154 | 2.975 | 49.581 | 49.581 |
| 2 | .944 | 15.725 | 97.879 | .944 | 15.725 | 97.879 | 2.898 | 48.298 | 97.879 |
| 3 | .065 | 1.083 | 98.963 | | | | | | |
| 4 | .041 | .684 | 99.647 | | | | | | |
| 5 | .020 | .336 | 99.983 | | | | | | |
| 6 | .001 | .017 | 100.000 | | | | | | |

Extraction Method: Principal Component Analysis

**Table 7. Components matrixa.**

| | Components | |
|---|---|---|
| | **1** | **2** |
| CAT | .955 | -.244 |
| Laccase | .949 | .294 |
| Amylase | .912 | .377 |
| SOD | .895 | -.436 |
| Cellulase | .862 | .482 |
| POD | -.860 | .482 |

Extraction Method: Principal Component Analysis

a. Two components have been extracted.

**Table 8. Rotation components matrixa.**

| | Components | |
|---|---|---|
| | **1** | **2** |
| Cellulase | .953 | .259 |
| Amylase | .915 | .369 |
| Laccase | .883 | .455 |
| POD | -.276 | -.947 |
| SOD | .334 | .938 |
| CAT | .511 | .843 |

Extraction Method: Principal Component Analysis Rotation Method: Varimax with Kaiser Normalization

a. Rotation converged in 3 iterations.

**Table 9. The scores of principal components and enzyme activity level.**

| Treament | $X_1$ | $X_2$ | $X_3$ | $X_4$ | $X_5$ | $X_6$ | $Z_1$ | $Z_2$ | K | Scores |
|---|---|---|---|---|---|---|---|---|---|---|
| $Zn^{2+}(0)$ | 0.167 | 0.323 | 0.144 | 0.970 | -1.260 | 1.122 | -0.667 | 1.31 | -0.35 | 56.5 |
| $Zn^{2+}(200)$ | 1.433 | 1.375 | 1.463 | 0.690 | -0.283 | 0.412 | 3.573 | -0.143 | 2.977 | 89.8 |
| $Zn^{2+}(400)$ | -0.492 | -0.555 | -0.682 | -0.199 | 0.201 | -0.094 | -1.457 | 0.087 | -1.207 | 47.9 |
| $Zn^{2+}(800)$ | -1.107 | -1.143 | -0.925 | -1.461 | 1.343 | -1.441 | -1.45 | -1.247 | -1.42 | 45.8 |

analysis aligned with the correlation results. amylase, laccase, and cellulase primarily break down carbohydrates during the growth of edible fungi, while superoxide dismutase and catalase mainly influence the antioxidant capacity of edible fungi. The findings indicated that a concentration of 200mg/L of ZnSO4 affected the growth and metabolism of *Pholiota adiposa* by increasing the activities of amylase, laccase, cellulase, and antioxidant enzymes. This suggests that ZnSO4 not only impacts the growth process but also the antioxidant process, with an optimal concentration.

The data analysis of the metabolic group revealed a positive correlation between zinc sulfate concentration and the concentrations of certain amino acids and organic acids such as glycine and D-gluconic acid.The joint analysis of enzymology and the metabolic group indicates that cellulase can utilize D-gluconic acid as a substrate, resulting in its degradation into useful products.This reaction provides organisms with energy and carbon sources, thereby promoting their growth and metabolism.Cellulase secretion is reduced at higher concentrations of

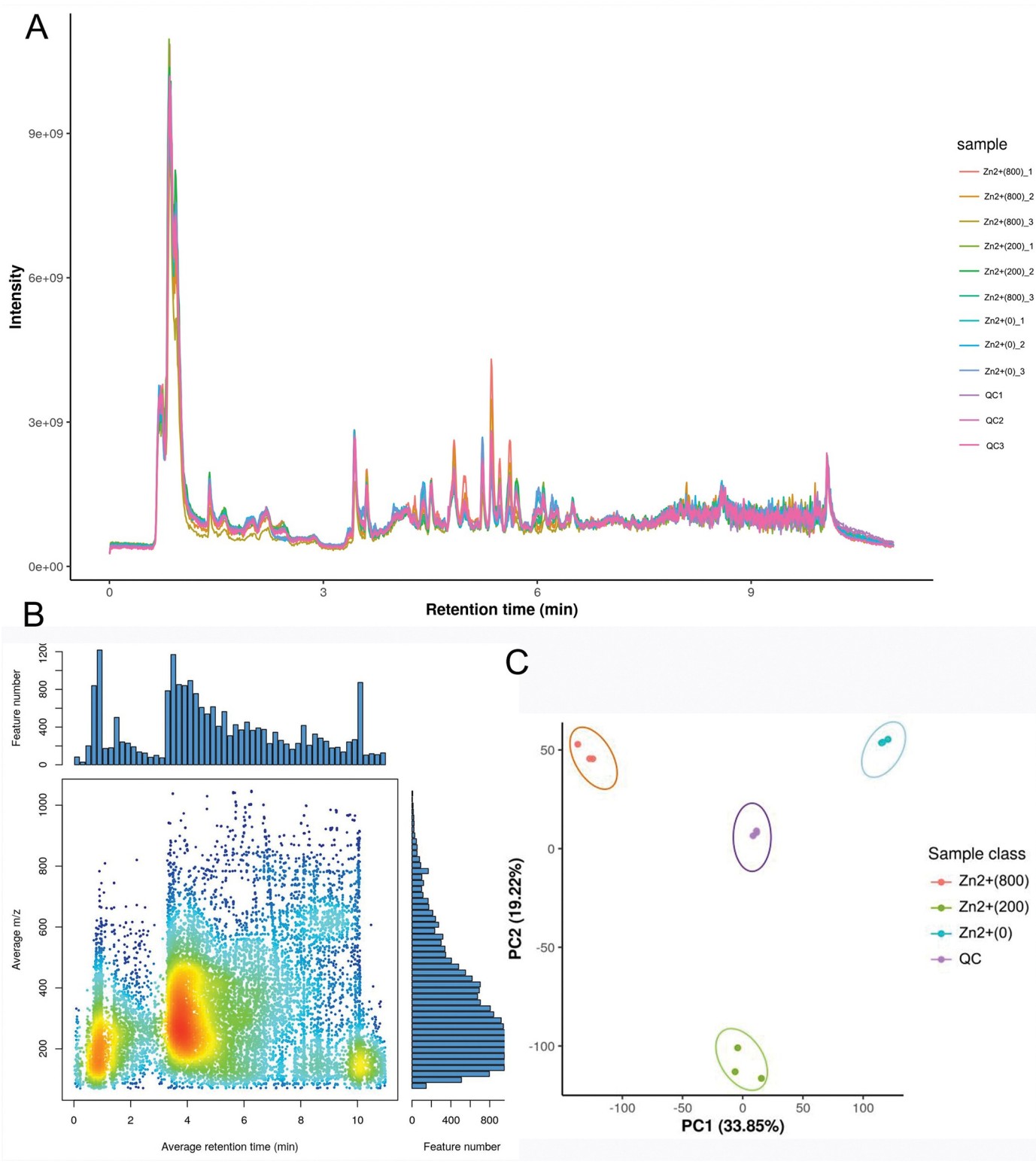

**Fig 11. Metabonomics data quality.**

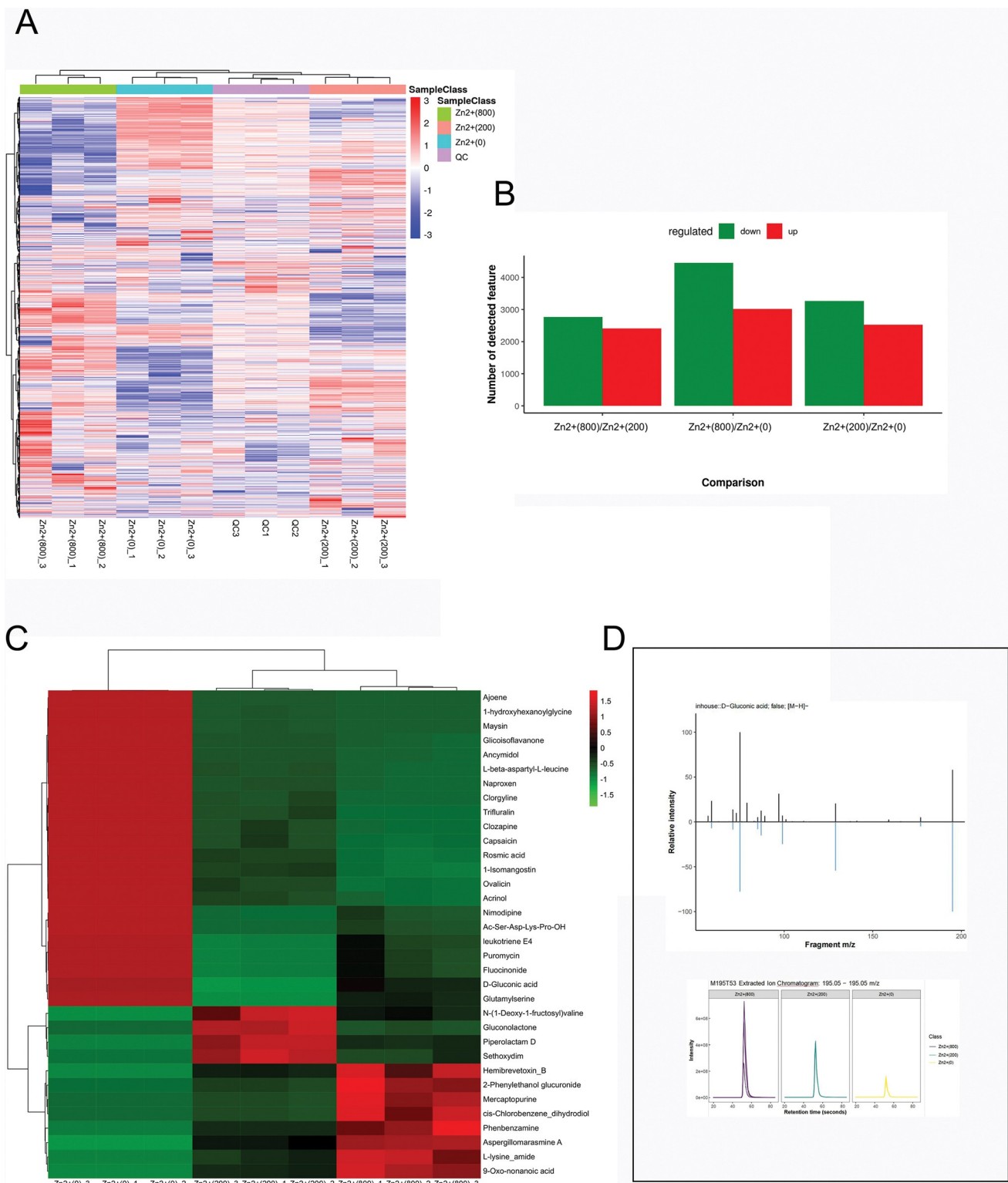

**Fig 12.** Analysis of Metabolites of *Pholiota adiposa* in Mycelium Growth by ZnSO₄ (A: Heatmap generated with the Zn regulated metabolites or substances in *Pholiota adiposa*; B: Effects of different concentrations of Zinc on the expression of metabolites in *Pholiota adiposa*; C: The heat map of Significantly differentially expressed metabolite; D: The metabolism spectrum of D-gluconic acid).

zinc sulfate.Therefore, when zinc sulfate is highly concentrated, the increased concentration of D-gluconic acid alone cannot provide sufficient energy and carbon sources for the mycelium, resulting in the inhibition of its growth due to insufficient cellulase.

Our study demonstrates that the addition of zinc enhances the enzyme activity and promotes metabolite expression in a concentration-dependent manner, suggesting the presence of an optimal concentration. It is worth noting the regulatory role of zinc in the growth regulation of fungi. These findings hold profound significance for the cultivation and production of edible fungi. Subsequent investigations should focus on evaluating the impact of zinc sulfate on the activity of specific enzymes in edible fungi, aiming to elucidate the potential underlying mechanisms. Furthermore, this study should investigate the duration of zinc supplementation and examine whether these effects differ among various fungal types or strains.

## Conclusion

Our study demonstrates that the addition of zinc enhances the enzyme activity and promotes metabolite expression in a concentration-dependent manner, suggesting the presence of an optimal concentration.

## Supporting information

**S1 Data.**
(ZIP)

**S2 Data.**
(ZIP)

## Author Contributions

**Supervision:** Peng Zhang.

**Writing – original draft:** Xiao-ying Ma, Peng Zhang.

**Writing – review & editing:** Tao Yang, Jun Xiao, Peng Zhang.

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
