## [Decision Letter · Decision Letter 0]

10 Oct 2023

PONE-D-23-26154The Effects of Zinc Sulfate on mycelial enzyme activity and metabolites of Pholiota adiposaPLOS ONE

Dear Dr.ZHANG,

Thank you for submitting your manuscript to PLOS ONE. After careful consideration, we feel that it has merit but does not fully meet PLOS ONE’s publication criteria as it currently stands. Therefore, we invite you to submit a revised version of the manuscript that addresses the points raised during the review process.

Please include the following items when submitting your revised manuscript:A rebuttal letter that responds to each point raised by the academic editor and reviewer(s). You should upload this letter as a separate file labeled 'Response to Reviewers'.A marked-up copy of your manuscript that highlights changes made to the original version. You should upload this as a separate file labeled 'Revised Manuscript with Track Changes'.An unmarked version of your revised paper without tracked changes. You should upload this as a separate file labeled 'Manuscript'.

We look forward to receiving your revised manuscript.

Kind regards,

Hasan Sardar, Ph.D.

Academic Editor

PLOS ONE

Journal Requirements:

Whilst you may use any professional scientific editing service of your choice, PLOS has partnered with both American Journal Experts (AJE) and Editage to provide discounted services to PLOS authors. Both organizations have experience helping authors meet PLOS guidelines and can provide language editing, translation, manuscript formatting, and figure formatting to ensure your manuscript meets our submission guidelines. To take advantage of our partnership with AJE, visit the AJE website (http://aje.com/go/plos) for a 15% discount off AJE services. To take advantage of our partnership with Editage, visit the Editage website (www.editage.com) and enter referral code PLOSEDIT for a 15% discount off Editage services. If the PLOS editorial team finds any language issues in text that either AJE or Editage has edited, the service provider will re-edit the text for free.

Reviewers' comments:

Reviewer's Responses to Questions

**Comments to the Author**

1. Is the manuscript technically sound, and do the data support the conclusions?

Reviewer #1: Yes

Reviewer #2: Partly

2. Has the statistical analysis been performed appropriately and rigorously? 

Reviewer #1: Yes

Reviewer #2: Yes

3. Have the authors made all data underlying the findings in their manuscript fully available?

Reviewer #1: Yes

Reviewer #2: Yes

4. Is the manuscript presented in an intelligible fashion and written in standard English?

Reviewer #1: Yes

Reviewer #2: No

5. Review Comments to the Author

Reviewer #1: Pholiota adiposa, belonging to the Fungi genus, is a widely distributed edible fungus with significant industrial importance. But only question is that why do we need to treat the edible fungi with Zinc sulfate? The article is written well and presented properly so, may be considered.

Reviewer #2: 1. Please ensure all plants' scientific names are written appropriately ie. in italics font

2. In the introduction - authors introduced about Pholiota adiposa, but later went on to highlight about Pleurotus ostreatus. Was it an unintentional typo error or could it be the way the sentence is written?

3. Please re-check and ensure your figures and tables are numbered correctly and matches the description / discussion that follows.

for example - There are 2 figures labelled as 11A, B,C and D. Which one is the actual Fig 11?

It was also difficult to follow the discussion referred by table 3, table 4, Fig 7 as the numbering was not properly matched.

4. For part "preparation of samples" - please explain as to why the need to grind the centrifuge tubes wall, shouldn't it be the sample?

5. For 'metabonomic sample processing' - are you using buffer or just distilled water? Please remove the word buffer if it's only distilled water. And should it be buffer, please explain the type and preparation of the buffer.

6. Figure 5 - Heat map. Please provide a small explanation for this interesting map.

7. Correlation analysis - based on table 2, your correlation description were unmatched. Please relook at what have been mentioned for amylase, cellulase, SOD and CAT.

8. Conclusion - please revise as based on the results, the effect of zinc is not in concentration dependent manner. It is instead Zn 200>0>400>800, clearly not a concentration dependent pattern.

9. Please do a simple language check e.g on page 14 'to explaining' suppose to be 'to explain' and on page 10 'maximum inject time' should be written as 'maximum injection time'

6. PLOS authors have the option to publish the peer review history of their article (what does this mean?). If published, this will include your full peer review and any attached files.

Reviewer #1: **Yes: **Arup Kumar Mukherjee

Reviewer #2: No

---

## [Author Response · Author response to Decision Letter 0]

17 Oct 2023

Reviewer #1: Pholiota adiposa, belonging to the Fungi genus, is a widely distributed edible fungus with significant industrial importance. But only question is that why do we need to treat the edible fungi with Zinc sulfate? The article is written well and presented properly so, may be considered.

Author : Thank you for your reviews!

Zinc is an essential micronutrient for plants and fungi. It plays a crucial role in enzymatic activities and helps in various metabolic processes. Treating fungi with zinc sulfate ensures that they have an adequate supply of this nutrient, which can promote their growth and development; Zinc sulfate treatment can help prevent certain diseases in edible fungi. Zinc has been shown to have antifungal properties, inhibiting the growth of harmful fungi that can cause diseases in mushrooms. By treating the fungi with zinc sulfate, the risk of infections and diseases can be reduced; Enhancing flavor and quality: Zinc sulfate treatment has been found to improve the flavor and quality of edible fungi. Zinc is involved in the synthesis of various compounds that contribute to the taste and aroma of mushrooms. By providing zinc to the fungi through zinc sulfate treatment, the resulting mushrooms can have a more desirable flavor and overall better quality; Increased yield: Zinc sulfate treatment has been shown to increase the yield of edible fungi. Zinc is involved in various metabolic processes, including photosynthesis and carbohydrate metabolism. By ensuring that the fungi have sufficient zinc, their overall growth and productivity can be enhanced, leading to higher yields.

Reviewer #2: 1. Please ensure all plants' scientific names are written appropriately ie. in italics font

Author : Thank you for your reviews! I have checked the full manuscript and modified it accordingly.

2.In the introduction - authors introduced about Pholiota adiposa, but later went on to highlight about Pleurotus ostreatus. Was it an unintentional typo error or could it be the way the sentence is written?

Author : What appears in the first paragraph of the introduction should be Pholiota adiposa. A noun error has occurred and has been revised.

3. Please re-check and ensure your figures and tables are numbered correctly and matches the description / discussion that follows.

for example - There are 2 figures labelled as 11A, B,C and D. Which one is the actual Fig 11?

It was also difficult to follow the discussion referred by table 3, table 4, Fig 7 as the numbering was not properly matched.

Author :Thank you for your reviews! The figures has been re-tagged, and there is no problem with the marking of figure 7 and tables 3 and 4.

4.For part "preparation of samples" - please explain as to why the need to grind the centrifuge tubes wall, shouldn't it be the sample?

Author :I am very sorry that my expression has been revised in the article, please review!

5.For 'metabonomic sample processing' - are you using buffer or just distilled water? Please remove the word buffer if it's only distilled water. And should it be buffer, please explain the type and preparation of the buffer.

Author : Thank you for your reviews! Buffer has been deleted.

6.Figure 5 - Heat map. Please provide a small explanation for this interesting map.

Author :Thank you for your reviews! The heat map is explained after ( figure 5) in ‘Enzyme activity determination’

7.Correlation analysis - based on table 2, your correlation description were unmatched. Please relook at what have been mentioned for amylase, cellulase, SOD and CAT.

Author :Thank you for your reviews! Correlation analysis has been recalibrated

8.Conclusion - please revise as based on the results, the effect of zinc is not in concentration dependent manner. It is instead Zn 200>0>400>800, clearly not a concentration dependent pattern.

Author :Thank you for your reviews! The conclusion has been revised.

9. Please do a simple language check e.g on page 14 'to explaining' suppose to be 'to explain' and on page 10 'maximum inject time' should be written as 'maximum injection time'

Author :Thank you for your reviews! I have modified it as required.

---

## [Editor Report · Decision Letter 1]

23 Nov 2023

The Effects of Zinc Sulfate on mycelial enzyme activity and metabolites of Pholiota adiposa

PONE-D-23-26154R1

Dear Dr. Zhang,

We’re pleased to inform you that your manuscript has been judged scientifically suitable for publication and will be formally accepted for publication once it meets all outstanding technical requirements.

Kind regards,

Hasan Sardar, Ph.D.

Academic Editor

PLOS ONE
---

## [Editor Report · Acceptance letter]

4 Dec 2023

PONE-D-23-26154R1 

The Effects of Zinc Sulfate on mycelial enzyme activity and metabolites of *Pholiota adiposa*

Dear Dr. Zhang:

I'm pleased to inform you that your manuscript has been deemed suitable for publication in PLOS ONE. Congratulations! Your manuscript is now with our production department. 

Kind regards, 

on behalf of

Dr. Hasan Sardar 

Academic Editor

PLOS ONE